# Mixed Polymeric Micelles for Rapamycin Skin Delivery

**DOI:** 10.3390/pharmaceutics14030569

**Published:** 2022-03-04

**Authors:** Guillaume Le Guyader, Bernard Do, Ivo B. Rietveld, Pascale Coric, Serge Bouaziz, Jean-Michel Guigner, Philippe-Henri Secretan, Karine Andrieux, Muriel Paul

**Affiliations:** 1Assistance Publique-Hôpitaux de Paris, Hôpitaux Universitaires Henri Mondor, F-94010 Créteil, France; guillaume.leguyader@aphp.fr (G.L.G.); muriel.paul@aphp.fr (M.P.); 2Centre Hospitalier Intercommunal de Créteil, F-94010 Créteil, France; 3Matériaux et Santé, Université Paris-Saclay, 92296 Châtenay-Malabry, France; philippe-henri.secretan@universite-paris-saclay.fr; 4SMS Laboratory (EA 3233), Université de Rouen-Normandie, Place Émile Blondel, 76821 Mont Saint Aignan, France; ivo.rietveld@parisdescartes.fr; 5Faculté de Pharmacie, Université de Paris, 4 Avenue de l’Observatoire, 75006 Paris, France; 6UMR 8038 CiTCoM, CNRS, University of Paris, 75006 Paris, France; pascale.coric@parisdescartes.fr (P.C.); serge.bouaziz@parisdescartes.fr (S.B.); 7Institut de Minéralogie, de Physique des Matériaux et de Cosmochimie (IMPMC), UMR CNRS 7590, MNHN, IRD UR 206, Université Sorbonne Paris Cité, F-75005 Paris, France; jean-michel.guigner@upmc.fr; 8UMR CNRS 8258—U1267 Inserm, Université de Paris, F-75006 Paris, France; karine.andrieux@parisdescartes.fr; 9EpidermE, Université Paris Est Créteil, F-94010 Créteil, France

**Keywords:** rapamycin, facial angiofibromas, mixed polymeric micelles, micelle-based hydrogel, stability, skin permeation

## Abstract

Facial angiofibromas (FA) are one of the most obvious cutaneous manifestations of tuberous sclerosis complex. Topical rapamycin for angiofibromas has been reported as a promising treatment. Several types of vehicles have been used hitherto, but polymeric micelles and especially those made of d-α-tocopherol polyethylene glycol 1000 succinate (TPGS) seem to have shown better skin bioavailability of rapamycin than the so far commonly used ointments. To better understand the influence of polymeric micelles on the behavior of rapamycin, we explored it through mixed polymeric micelles combining TPGS and poloxamer, evaluating stability and skin bioavailability to define an optimized formulation to effectively treat FA. Our studies have shown that TPGS improves the physicochemical behavior of rapamycin, i.e., its solubility and stability, due to a strong inclusion in micelles, while poloxamer P123 has a more significant influence on skin bioavailability. Accordingly, we formulated mixed-micelle hydrogels containing 0.1% rapamycin, and the optimized formulation was found to be stable for up to 3 months at 2–8 °C. In addition, compared to hydroalcoholic gel formulations, the studied system allows for better biodistribution on human skin.

## 1. Introduction

Tuberous sclerosis complex (TSC) is an autosomal dominant disorder, characterized by formation of hamartomas in various organs [1]. TSC results from a mutation in the TSC1 or TSC2 gene, leading to mammalian target of rapamycin (mTOR) overactivation and continued cell proliferation [2]. Antagonism of the mTOR pathway, such as with rapamycin, offers new treatment options for patients with TSC. Among the associated clinical signs, facial angiofibromas (FA) are one of the most obvious cutaneous manifestations of TSC, which can be disfiguring, hence causing substantial psychological distress with a major impact on quality of life [3,4]. Invasive procedural treatments such as surgery or laser involve pain and hypertrophic scars and are not entirely satisfactory due to the recurrence of skin lesions [4,5]. For this reason, medical treatment with topical rapamycin has so far been a particularly studied option. In the absence of a marketed pharmaceutical product for the topical route, various in-house formulations of rapamycin have been developed and clinically tested. They appear effective and safe for the management of facial angiofibromas (FA) [6,7,8,9,10]. However, the efficacy endpoints, duration of treatment, types of formulations and concentrations tested were heterogeneous, making it difficult to compare these studies [6]. A study has identified different formulations to compare their performance in terms of permeation [11]. The results showed the influence of the vehicle and the thermodynamic activity on the cutaneous bioavailability of rapamycin. Hydrogel vehicles, due to high alcohol content used to dissolve slightly soluble rapamycin, produced a better release than the lipophilic semi-solid base. However, the disadvantage of ethanol is that it induces dryness and irritation of the skin [9,12,13].

Nanocarriers such as liposomes, micelles and nanoparticles have been used to improve skin permeation of poorly permeable drug substances. They also improve other criteria such as higher stability, a larger surface area in contact with the skin, less toxicity and greater skin protection [14,15,16,17]. Encapsulated, the drug can penetrate the hair follicle and the skin, be released in a prolonged manner, interact better with the lipid transport of the skin, and offer better distribution of hydrophilic substances in the stratum corneum [14,18,19]. Due to their high solubility, high loading capacity and controllable release with a long duration of action, biodegradable polymeric micelles have gained considerable popularity [18,20,21,22]. Polymeric micelles are formed from copolymers, each consisting of hydrophilic and hydrophobic monomers, which self-assemble when the critical micelle concentration (CMC) is exceeded to form a specific core-corona structure. Their advantages lie in their very low CMC (about 1000 times lower than known values for low molecular weight surfactants), which allows them to be used in small quantities while ensuring good stability in aqueous media [23,24]. Using tocopherol polyethylene glycol 1000 succinate (TPGS)-based micelles, Quartier et al. have developed a rapamycin micellar hydrogel for the treatment of facial angiofibromas. This system, stable for at least 3 months at 4 °C, improved the cutaneous bioavailability of rapamycin in the stratum corneum and viable epidermis compared to controls (0.2% rapamycin dispersed in vaseline and paraffin liquid) [16]. However, as TPGS is also widely used as a cosurfactant for mixed micelles and exhibits synergistic effects with other polymers, such as poloxamer F127 or P123, we sought to assess whether such a combination could give rise to better characteristics in terms of stability and skin permeation. So, the aim of the present work was to study the relevance of rapamycin-loaded mixed micelles as an interesting approach for cutaneous rapamycin delivery.

## 2. Materials and Methods

### 2.1. Polymeric Micelles and Hydrogel Preparation

#### 2.1.1. Preparation of Rapamycin Micelle Solution

Rapamycin polymeric micelles were prepared by the thin film hydration method. Rapamycin and copolymers were dissolved in ethanol (Cooper, Melun, France). Ethanol was subsequently evaporated at 40 °C by using a rotary evaporator (BuchiTM Rotavapor, R-114, Zurich, Switzerland) to obtain a thin and dried residue. The dried residue was hydrated with purified water (Laboratoire Aguettant, Lyon, Ftrance) to obtain a mixture containing 0.1% (*w*/*v*) of rapamycin (Inresa, Bartenheim, France). Addition of water spontaneously generated rapamycin-loaded nanomicelles. After equilibration overnight at 2–8 °C, this mixture was filtered through a 0.22-μm-polyvinylidene-fluoride (PVDF; Sigma-Aldrich, St. Louis, MO, USA) filter to remove unentrapped drug and aggregates and other foreign particulates.

#### 2.1.2. Preparation of Polymer Micelle-Based Hydrogel Containing Rapamycin

To feature adequate rheological properties to facilitate skin application, rapamycin micelle solutions were jellified by incorporating thickening agent (i.e., Carbopol^®^ 974P; Cooper, Melun, France) to achieve a final concentration of carbomer of 1% *w*/*v*. Carbopol^®^ was previously neutralized by drop-wise addition of sodium hydroxide (1N) (Merck, Darmstadt, Germany).

### 2.2. HPLC for Rapamycin and Seco Rapamycin Determination in Micelles

High performance liquid chromatography (HPLC) for rapamycin and seco rapamycin determination was based on a Dionex Ultimate 3000 coupled with a Photodiode Array Detector PDA-3000 (Thermo-Fisher-Scientific, Ulis, France). An Interchim 250 × 4.6 mm KR-C18 (5 µm) column (Interchim, Clichy, France) was used and maintained at 40 °C during analysis. Elution was performed in a gradient, with the ratios of mobile phases A and B changing from 50:50 to 10:90 over 25 min. Mobile phase A consisted of 0.003% formic acid (Merck, Darmstadt, Germany) in purified water, while mobile phase B consisted of 0.003% formic acid in acetonitrile (Merck, Darmstadt, Germany). The flow rate was set at 1.0 mL∙min^−1^, and the injection volume was 50 µL. UV detection was set at 277 nm. The HPLC method was validated by rapamycin assay based on the International Council for Harmonisation (ICH) guidelines [25]. In such conditions, secorapamycin and rapamycin were eluted in about 18.5 min and 20.0 min, respectively.

### 2.3. Micelle Solution: Influence of the Polymeric Composition on Different Parameters Monitored

#### 2.3.1. Design of Experiments to Assess the Influence of the Type of Polymer Used

A two-factor full factorial design was employed to statically analyze the influence of the two polymers (i.e., poloxamer and TPGS) and to obtain an optimized rapamycin polymeric micelle composition using Design-Expert^®^ software (version 12, Stat-Ease Inc., Minneapolis, MN, USA). Nine formulae were prepared in six replicates, giving a total of 54 runs. Different weight ratios of TPGS were mixed with either poloxamer F127 (BASF, Geismar, LA, USA) or P123 (Sigma-Aldrich, St. Louis, MO, USA). Thus, the independent variables were: poloxamer type (A) and weight ratio of TPGS to total copolymer in nanomicellar mixture (B). The composition of the investigated formulae is shown in Table 1. During the screening studies, polymer content was kept constant at 25 mg∙mL^−1^. The responses selected to achieve optimized micelle formulation were maximize entrapment efficiency (Y_1_), minimize micellar size (Y_2_), minimize polydispersity index (Y_3_), zeta potential (Y_4_), maximize thermodynamic stability (Y_5_) and maximize in vitro flux (Y_6_), as shown in Table 1. Analysis of variance (ANOVA) was carried out to estimate the significance of the model. Probability *p*-values (*p* < 0.05) denoted significance.

#### 2.3.2. DoE to Assess the Influence of the Polymer Amount (Drug-to-Polymer Ratio)

Additional experiments were performed to investigate the impact of the polymer-to-drug ratio on the physicochemical properties of the optimal formulation, which had been previously determined using the two-factor full factorial design. Thus, the total polymer concentration was tested at concentrations of 2.5, 5, 10 and 25 mg∙mL^−1^, while keeping the rapamycin concentration constant. Preparations were compared in terms of drug entrapment efficiency (EE%, see below), size, morphology, and in vitro flux.

#### 2.3.3. Morphology, Size, Polydispersity Index and Zeta Potential Determination

Micelle morphologies were determined from cryo-transmission electron microscopy (cryo-TEM) images. A drop of micelle solution was deposited on Lacey carbon membrane grids. The excess of liquid on the grid was absorbed with a filter paper, and the grid was quench-frozen quickly in liquid ethane to form a thin vitreous ice film. Once placed in a Gatan 626 cryo-holder cooled with liquid nitrogen, the samples were transferred to the microscope and observed at low temperature (−180 °C). Cryo-TEM images were recorded on an Ultrascan 1000, 2k × 2k CCD camera (Gatan, Pleasanton, CA, USA), using a LaB6 JEOL JEM2100 (Jeol, Tokyo, Japan) cryo microscope operating at 200 kV with a JEOL low-dose system (Minimum Dose System, MDS) to protect the thin ice film from any irradiation before imaging and reduce the irradiation during the image capture.

The hydrodynamic diameter and size distribution (polydispersity index, PDI) of the RPN solutions were measured with the dynamic light scattering (DLS) method using a Zetasizer Nano ZS (Malvern Instruments, Worcestershire, UK) at 25 ± 0.1 °C with angle of 173°. Zeta potential was measured by the electrophoretic mobility method with the same apparatus.

#### 2.3.4. Drug Entrapment Efficiency (EE%) and Drug Loading (DL%) Determination

Entrapment efficiency of rapamycin in polymeric micelles was determined after separation of the unincorporated drug by filtration through 0.22 μm PVDF filter membrane. Then, clear supernatant was diluted with methanol to ensure the complete destruction of the micelles and the release of loaded rapamycin. The amount of entrapped rapamycin was quantified by UV-HPLC. The EE% and DL% were then calculated according to the following equations:EE% = (Weight of drug into micelles)/(Weight of the feeding drug) × 100(1)
DL% = (Weight of drug into micelles)/(Weight of the feeding drug and polymers) × 100(2)

#### 2.3.5. Study of Drug–Polymer Interactions/Inclusion by Attenuated Total Reflectance-Fourier Transform Infrared Spectroscopy (ATR-FTIR) and Nuclear Magnetic Resonance (NMR)

Analyses were performed on free rapamycin merely mixed with polymer and rapamycin-loaded micelles obtained by the thin-film hydration method (see Section 2.1). In both cases, samples were prepared/dispersed with deuterated oxide solvent (D_2_O) to limit water –OH stretch vibration.

##### Fourier Transform Infrared Spectroscopy (FTIR)

The ATR-FTIR was recorded against the background by using a universal ATR sampling assembly (Spectrum-Two; Perkin-Elmer, Waltham, MA, USA). For each sample, 25 scans were obtained at a resolution of 4 cm^−1^ in the range of 4000–400 cm^−1^ at room temperature.

##### Nuclear Magnetic Resonance (NMR)

NMR experiments were recorded at 20 °C on an Avance Bruker spectrometer (Bruker, Billerica, MA, USA) operating at 600.13 MHz and equipped with a cryoprobe. One–dimensional spectra were obtained with 2048 points and a spectral width of 7002.8 Hz. The transmitter frequency was set to the water signal, and the solvent resonance was suppressed by using excitation sculpting with pulsed-field gradients [26]. All data were processed using Bruker Topspin (v 1.5pl7) software (Bruker Biospin, Ettlingen, Germany). Temperature was controlled externally using a special temperature control system (BCU 0.5; Bruker, Billerica, MA, USA). Data were zero filled, and π/6 phase-shifted sine bell window function was applied prior to Fourier transformation.

#### 2.3.6. In Vitro Flux Determination

Permeation of the drug was studied through a Strat-M^®^ synthetic membrane (Merck Millipore, Burlington, MA, USA), using the standardized methodology of Franz diffusion cells. These had an effective diffusion area of 1.767 cm^2^ and a receptor volume of 7.0 mL (Teledyne Hanson^®^ Research, Chatsworth, CA, USA). The receptor compartment was set at 32 °C, using a circulating water bath, to mimic the skin temperature at physiological level and stirred at a speed of 400 rpm. The receptor compartment was filled with 30:70 (*v*:*v*) filtered ethanol:phosphate buffer pH 7.4 solution (PBS, Dako^®^, Carpinteria, CA, USA). The diffusion cells were allowed to equilibrate at 32 °C for 30 min. Then, at time zero, 0.3 g of preparation was added to the donor compartment of each Franz diffusion cell. At pre-determined time intervals, 500 µL of receptor fluid was collected and the same volume of fresh preheated medium was reintroduced into the receiver to retain the sink conditions in the system. Samples were analyzed using the HPLC method to determine the amount of rapamycin diffused. The steady state flux at 24 h, in vitro skin permeation steady state flux (Jss) (µg cm^−2^ h^−1^), was determined by measuring the slope from the plot of the cumulative amount permeated versus time [11].

#### 2.3.7. Ex Vivo Permeation Study on Micelle Hydrogel

The optimized formulation was applied on dermatomed human skin obtained from human volunteers followed ethical principles and was previously approved by French Ethics Committee (N◦AC 2014-2233 the 18 February 2015 and DC 2014-2227 the 13 January 2015). Dermatomed human skin (thickness 500 µm) from abdominoplasty was purchased from Proviskin (Besancon, France). All of the experimental conditions of the in vitro permeation study were maintained except for the receiver chamber, which was filled with filtered PBS solution pH 7.4 without ethanol. At the end of the permeation experiment, the diffusion cells were dismantled, and each sample was carefully washed to ensure the removal of the residual formulation from the skin surface. Skin samples were dried with a cotton swab. To verify the effectiveness of the wash procedure, the rapamycin remaining on the skin was quantified by HPLC and was found to be below the limit of quantification of the analytical method. The epidermis was subsequently separated from the dermis, and each compartment was cut into small pieces and soaked in methanol for 12 h with continuous stirring at room temperature. The extraction procedure was validated in our previous study (data not shown) [11]. The extraction skin samples were then centrifuged at 10,000 rpm for 15 min and analyzed by HPLC.

### 2.4. Accelerated and Long-Term Stability Studies of Micelle Solution and Hydrogel

Stability of the micelle solutions was evaluated after storage at 2–8 °C for up to 6 months, or at 40 °C/75% residual humidity for up to 7 days. As for the hydrogel, stability was monitored for up to 3 months at 2–8 °C. Each sample was previously filtered through a 0.22 µm PVDF filter membrane, and the amount of entrapped rapamycin (EE%) was determined. The average value was expressed as a percentage relative to the initial value before storage in the chambers.

A micelle-based hydrogel was prepared in triplicate (*n* = 3) from the optimal micelle formulation and characterized in terms of appearance (color, transparency), pH, rapamycin content, rheological properties, and stability. Hydrogels were packaged in aluminum tubes (Cooper, France) and stored as such for the stability studies. For each run, samples were withdrawn from the tubes, and appearance (by visual inspection and optical microscopy), rapamycin EE%, rheology and pH were determined. pH was measured using a calibrated pH-meter Consort-P901 (Cole Parmer, Wertheim, Germany). Optical microscopy utilized an Olympus^IM^ (Olympus, Tokyo, Japan) coupled to a Sony camera XCD-U100CR (Sony Corporation, Tokyo, Japan). Rheological experiments were conducted at 25 °C using rheometer RM-200 CP-4000 Plus (Lamy Rheology Instruments, Champagne-au-Mont-d’Or, France) equipped with cone-plate geometry CP-2020 (with diameter 20 mm, cone angle 2°, truncation 50 µm). The speed setting was adjusted over a range of 10 to 150 s^−1^ for 180 s. The evolution of various parameters was compared over time, between formulations and with respect to time zero. In case these parameters were all equivalent at T0, relative differences between the formulations helped identify the best polymer composition among those tested.

## 3. Results

### 3.1. Micelle Optimization and Characterization

#### 3.1.1. Spectral Highlights of Rapamycin-Loading Micelles

The inclusion of rapamycin in micelles was demonstrated by ATR-FTIR and NMR spectroscopy.

The ATR-FTIR spectrum of a non-loaded rapamycin mixture prepared by extemporaneously dispersing/solubilizing rapamycin and TPGS/poloxamer P123 (1/1 *w*/*w*) in D_2_O (red line) showed a broad but weakly intense peak stretching from 3200 to 3550 cm^−1^, corresponding to OH vibrations with intermolecularly engaged H (Figure 1 and Appendix A). In the case of rapamycin-loaded micelles, the same peak became much more intense, which would imply a strengthening of this phenomenon, probably due to stronger polar interactions between rapamycin and the hydrophilic parts of the polymers. Furthermore, the near disappearance of a cluster of peaks located between 2875 and 2932 cm^−1^, observed in the case of rapamycin-loaded micelles, most probably related to the C–CH_3_, O–CH_3_ and aliphatic C–H stretching vibrations and, characteristic of certain functions of the active substance, seemed to clearly indicate that the latter was deeply included in the inner part of the micelles, becoming non-visible to the FTIR detection. Likewise, the conjugated C=O and conjugated C=C stretching vibrations (1718 cm^−1^ and 1635 cm^−1^, respectively) were no longer present for the rapamycin-loaded micelles. In a nutshell, examination of the spectrum recorded for the rapamycin-loaded micelles yielded a completely different configuration, showing the disappearance of the peaks pertaining to rapamycin, implying its deep inclusion within micelles.

The same phenomenon was also observed in NMR (Figure 2). Chemical shifts related to rapamycin were no longer detected in the rapamycin-loaded micelles. More precisely, the characteristic chemical shifts of the hydrophobic hydrogens (C–H) disappeared, demonstrating the interaction of rapamycin with the hydrophobic parts of the micelles. Since the cores of the micelles dispersed in aqueous media were hydrophobic, it was easy to conclude that rapamycin was included in the inner part of the micelles featuring hydrophobic functions. This helped to explain the poor EE% observed with micelles consisting only or mostly of F127 (see below).

However, while we succeeded in showing the existence of an interaction between rapamycin and the hydrophobic parts of the polymers, these data did not allow us to know whether the micelles present were hybrid or mixed micelles, or if there was cohabitation between TPGS-micelles and P123-micelles. Mechanistically speaking, it is likely that due to the proximity of their hydrophilic lipophilic balance (HLBs), the joint use of P123 and TPGS would probably allow both systems to be present. On the other hand, when TPGS and F127 are mixed, the two polymers should rather, but not exclusively, afford two separate populations due to the significant differences in their HLBs. However, these assumptions were subsequently investigated by studies of micelle size and polydispersity index.

#### 3.1.2. Influence of Polymer Composition

As discussed in detail in this chapter, the results of the DoE confirmed the interest of working with mixed micelles, provided the right type of poloxamer is chosen.

##### On the EE%

We studied the impact of the ratio of the two copolymers and the type of poloxamer on micelle characteristics, such as size, rapamycin entrapment efficiency (%), zeta potential, and PDI, as well as on the thermodynamic stability of the mixture and the in vitro bioavailability. The corresponding results are gathered in Table 2. The influence of parameters on the controlled criteria was modeled. Of the models tested, only quadratic least squares models led to a difference between adjusted and predicted R^2^ of no more than 0.2 [27], and this was observed for almost all responses (ANOVA, *p* < 0.05) studied apart from zeta potential (ANOVA, *p* = 0.3525) (Table 2). A significant impact on most of the criteria studied implied the possibility of adjusting the polymer composition to obtain an optimum desired by the experimenter.

One of the criteria that has been extensively evaluated is EE%, potentially the source of good stability and efficacy of rapamycin. Based upon the data presented in Table 2, EE% was in all cases above 80%. These outcomes are like those obtained by Quartier et al. using TPGS-only micelles [16]. Although TPGS was shown to produce a statistically better EE% than either F127 or P123 (ANOVA, *p* < 0.05), the presence of one of the poloxamers tested (F127 or P123) alone appeared to almost match the performance of TPGS (Figure 3).

However, when the EE% was monitored after a certain storage time of the tested mixtures at 4 °C (6 months), it was found to be closely dependent on the polymer composition of the micelles (Table 1 and Table 2; Figure 4). The stability of EE%, which in a way reflected the affinity of rapamycin for the polymers, was indeed overall ensured by TPGS and/or P123 (Table 1 and Table 2; Figure 4). It seems, then, that the partitioning of rapamycin between the micelles and the supernatant was promoted with the decrease in storage temperature. The corollary of this is that accelerated temperature conditions (40 °C) will be unfavorable to EE% and that heating seems to be a factor that accentuates certain phenomena not perceived at lower temperatures. Indeed, at 40 °C, the retention of rapamycin decreased more strongly in the presence of P123 than it did in the presence of TPGS, in all cases in the absence of F127 (7 days at 40 °C).

When the micelles consisted of F127, if the relative content of TPGS did not exceed 50%, the release of the active substance into the supernatant was considerable and may have corresponded to a loss in the micelles of most of the active substance (Table 1 and Table 2). This phenomenon could be seen over time even at 4 °C, which eventually made F127 not a good candidate. Poloxamers P123 and F127 are linear triblock copolymers of poly(ethylene oxide) (PEO) and poly(propylene oxide) (PPO), often denoted PEO-PPO-PEO. F127 is PEO-rich, while P123 is PPO-rich, giving them HLB values of 22 and 8, respectively [28]. It is likely, then, that the high hydrophilic proportion of F127 caused a progressive hydration of the micelles incompatible with the hydrophobic nature of rapamycin (log P 4.3), which led to its massive release over time. However, as seen above, the EE% stability is in favor of TPGS compared to P123 at elevated temperature, while TPGS has a higher HLB value (HLB 18 versus HLB 8). This is, therefore, a balance that needs to be struck to promote rapamycin entrapment, and it seems that TPGS is well-suited for this purpose [16,28,29].

Any decrease in EE% appeared to be inversely related to the appearance of secorapamycin, the major degradation product of rapamycin, detected in the supernatant (Figure 5). This is not surprising since polymeric micelles, those based on TPGS [16], have been shown to effectively protect rapamycin from chemical degradation. It is thus likely that a release of rapamycin outside the micelles seems to expose it to conditions favoring its degradation to secorapamycin.

##### On the Micelle Size and Polydispersity Index

Particle size was investigated, as it is generally considered that this parameter contributes significantly to the drug’s skin penetration. It has been established that the smaller the particle size, the better the passage of active substances into the deeper layers of the skin [30,31]. Besides that, the solubility of an active substance within its vehicle can be a contributing factor to this, and solubility is often inversely proportional to the size of the micelles, owing to the increase in the surface area to volume ratio [32].

The average hydrodynamic diameter of the polymeric micelles we studied ranged from 10.7 ± 0.1 nm to 29.7 ± 2.6 nm, which agrees with the results of other studies [16,32,33,34]. However, there seems to be a link between the hydrophobic part of the polymer studied and the size of the micelles [35]. Indeed, the micelles stemming from P123 (F1: 17.7 ± 0.5 nm) showed a statistically lower hydrodynamic diameter than those made with F127 (F6: 29.7 ± 2.6 nm) (Table 2; Figure 6). Moreover, in both cases, as TGPS mass ratios increased, micelle size decreased, probably due to enhanced hydrophobic interactions resulting in more compact structures, which may explain the increase in EE% values in both cases, as seen previously (Table 2).

If we look at the size variability of the studied micelles by determining their PDI, it appears that across the polymeric compositions described in Table 1, the PDI is narrow, ranging from 0.020 ± 0.011 to 0.158 ± 0.004 (Table 2). Such a distribution seems to be the consequence of a co-micellization between TPGS and poloxamer [36]. However, contrary to the case of P123, when the TPGS mass ratio is low (F6-25 and F7-25), the PDI of the F127-based systems is significantly higher (Table 2), thus suggesting the cohabitation of several populations, which corroborates and reinforces the hypotheses put forward previously.

Therefore, at this stage of the study, if the concept of TPGS-poloxamer co-micellization is to be generalized towards rapamycin or a hydrophobic molecule, a poloxamer with a low HLB should be chosen to better interact with TPGS and allow the formation of a small micelle size with a narrow PDI, as these characteristics seem to correlate with the stability of the EE%.

##### On the Zeta Potential

In addition to the above criteria, the evaluation of the zeta potential allowed us to estimate the risk of micelle aggregation. Regardless of the polymeric composition studied, the zeta potential ranged from −2.5 ± 1.5 to −0.4 ± 0.5 mV, which was not different from what has been published so far and where it has been shown that the risks of aggregation are low (Figure 6) [32,37,38]. In addition, it is likely that the absence of strong repulsive charges between the micelles and negatively charged cell membranes may induce diffusion through the skin via paracellular routes [39].

##### On Rapamycin In Vitro Permeation Using Franz Diffusion Cells

The objective of these studies was to identify the best performing polymer composition for in vitro permeation. In this case, in contrast to stability performance, it seemed that poloxamer played a more interesting role than TPGS (Figure 7). However, this only concerned P123. Indeed, as shown in Table 2, the highest flux (2.07 ± 0.13 µg cm^−2^ h^−1^), leading to highest cumulative amount at 24 h, was obtained with the P123-only micelles, which turned out to be about twice the values obtained with TPGS alone and F127 alone.

Although the combination with TPGS decreased the flux by about 10% for any TPGS mass ratio not exceeding 50%, the concomitant use of TPGS remained in consideration, as the latter favored the inclusion of rapamycin within the micelles for better stability.

This confirmed the interest of working with mixed micelles, provided the right type of poloxamer is chosen.

#### 3.1.3. Influence of Polymer Concentration

The previous studies were performed with 25 mg∙mL^−1^ of polymers in mixtures containing different types of micelles. In this case, we selected the 50/50 TPGS/P123 (F3) based mixture offering the best compromise in terms of performance to study more precisely how the system may be influenced by polymer concentration.

Overall, it is interesting to note that the EE% was comparable to the values presented in Table 2, and no trend was detected as a function of polymer concentration (Table 3). This means that the lowest polymer concentration tested (2.5 mg∙mL^−1^) was higher than the critical micellar concentration of one or both polymers used. Indeed, microscopic analysis by cryo-TEM showed spherical micelles formed in this concentration condition in the presence of rapamycin (Figure 8).

However, at the concentration of 2.5 mg∙mL^−1^, the average micelle size increased sixfold compared to the other concentrations tested, with an increased PDI, an indicator of greater system disorder (Table 3). Even if this phenomenon was not immediately correlated with a drop in EE%, as shown previously, it is highly likely that, despite the absence of tests demonstrating it, this drop would be visible over time and/or under accelerated conditions.

If we were able to determine the minimum concentration allowing us to obtain a stable micellar system based on the size of the micelles and the PDI, it was not possible through these two criteria and the EE% to identify the concentration from which the system would no longer meet the desired performance.

That is why we also monitored the diffusion of rapamycin through Strat-M^®^ membrane. Figure 9 describes the evolution of the in vitro steady-state flux as a function of the polymer concentration. In the concentration range 2.5–10 mg∙mL^−1^, the steady-state flux evolved in an almost linear way. This means that the polymer concentration promoted transmembrane passage of rapamycin [28,40], but only up to a certain value, since 25 mg∙mL^−1^ produced the same result as for 10 mg∙mL^−1^ (Figure 9). This could mean that above a certain concentration of polymer, i.e., micelles, the receiving medium is no longer able to extract more rapamycin likely due to a greater retention capacity exerted by a higher number of micelles.

However, although the optimal polymer concentration is around 10 mg∙mL^−1^, we preferred to continue working at 25 mg∙mL^−1^ to give maximum stability to the micellar system to cope with a change of environment due to the formulation of a hydrogel containing 0.1% rapamycin.

### 3.2. Characterization of the Selected Micellar Hydrogel

#### 3.2.1. Physicochemical Stability

A hydrogel formula containing 0.1% rapamycin was prepared based on TPGS-P123 mixed polymeric micelles. The objective was to allow the administration of rapamycin by the cutaneous route. The gel featured a smooth and homogeneous texture containing entirely dissolved rapamycin.

For at least 3 months at 4 °C in the dark, it was monitored to verify its stability in terms of assay, appearance to detect any precipitation, degradation product and rheological behavior. Up to 3 months, as shown in Figure 10, no significant decrease in rapamycin was noticed. Secorapamycin was detected, but beneath 1%. No change in the rheological behavior was observed, either (Figure 11).

This result is promising as it allows consideration for the possibility of hospital preparations to treat patients with angiofibromas.

#### 3.2.2. Ex Vivo Study

To assess the permeation of rapamycin micellar hydrogel, an ex vivo study on human skin was performed.

We monitored the cutaneous biodistribution of rapamycin in the different layers of human skin and compared the results to those of a hydroalcoholic gel, with the understanding that hydroalcoholic gels are currently the most widely used vehicle for administering rapamycin.

In addition, we also drew some parallels with Quartier et al. [16] data to determine whether the combination of TPGS with P123 merited further investigation.

Twenty-four hours after deposition on human skin, the cumulative presence of rapamycin in the epidermis was about 2.5 times higher than that measured for the hydroalcoholic gel (about 1900 versus 700 ng∙cm^−2^) (Figure 12). In the dermis though, the cumulative presence of active substance was comparable for the two types of gels tested (about 700–800 ng∙cm^−2^). This implies that after 24 h, the total cumulative amount of rapamycin in the epidermis and dermis combined was greater with the use of our polymer-based gel.

While remaining very cautious and comparing only what is comparable with published data [16], it appears that the concomitant use of P123 in equal parts with TPGS led to higher cumulative depositions than those achieved with TPGS alone. It is, therefore, likely that the addition of P123 would have improved the thermodynamic activity of the polymeric micelle-based carrier. However, we will later seek to complement and strengthen these initial assessments by testing a wider range of rapamycin concentrations and different TPGS/P123 ratios.

Based on these results and the fact that intercluster penetration (deposition between corneocytes and between corneocyte clusters) and/or within hair follicles have been suggested as the preferred [41,42,43], but not the only [44,45,46], transport routes for polymeric micelles, we hypothesized that a concentration of micelles at the epidermal level, within which rapamycin’s chemical stability is maintained, may act as a depot for the dermis, which, upon repeated administration, could enhance the concentration of rapamycin in the deeper layer through continuous release of rapamycin, resulting in the formation of a diffusion gradient. Nonetheless, further tests will have to be carried out to support these hypotheses.

## 4. Conclusions

Topical rapamycin has been shown to be effective in children. Adults, on the other hand, respond much less well to treatment due to a skin barrier that is a greater hurdle to rapamycin bioavailability. Therefore, the search for vehicles that can help improve the passage of the active substance is still ongoing.

Recent evidence for rapamycin describes the use of TPGS-based polymeric micelles dispersed in a gel with very promising results compared to, for example, ointment-type vehicles. We have worked in this direction by evaluating the interest of mixed micelles based on TGPS in combination with other poloxamer-type polymers, both present in low concentrations of no more than 2.5% *w*/*w* of polymers due to their very low CMC.

We then showed that the choice of poloxamer is important—preferably a poloxamer with a low HLB, i.e., with a high hydrophobic fraction, such as P123.

We conclude that while the poloxamer P123 does not further improve the stability of the system, it at least seems to play a greater role in the thermodynamic activity of the vehicle, improving the bioavailability of rapamycin.

## Figures and Tables

**Figure 1 pharmaceutics-14-00569-f001:**
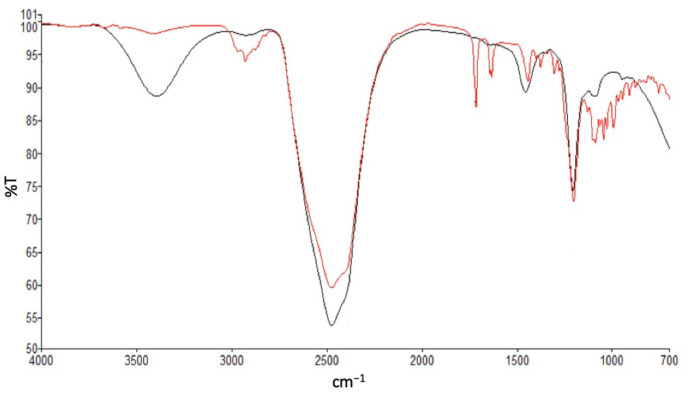
ATR-FTIR spectra of rapamycin-unloaded micelle (red line) and rapamycin-loaded micelle (black line) in deuterated water.

**Figure 2 pharmaceutics-14-00569-f002:**
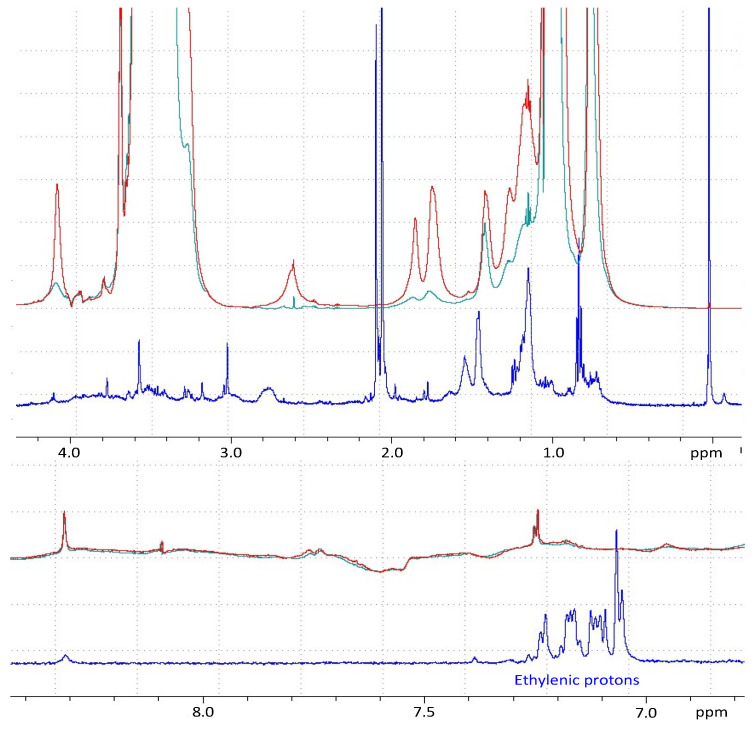
Nuclear magnetic resonance (NMR) spectra of rapamycin (blue line), drug-free micelles (red line) and rapamycin-loaded micelles (green line) in deuterated water.

**Figure 3 pharmaceutics-14-00569-f003:**
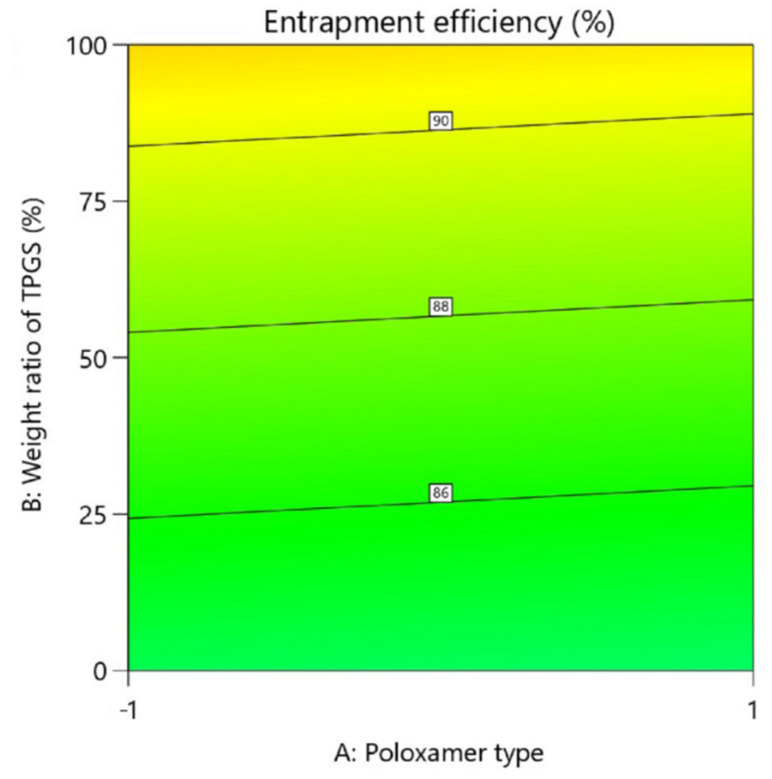
Contour plot showing relative effect of poloxamer type and weight ratio of TPGS on entrapment efficiency.

**Figure 4 pharmaceutics-14-00569-f004:**
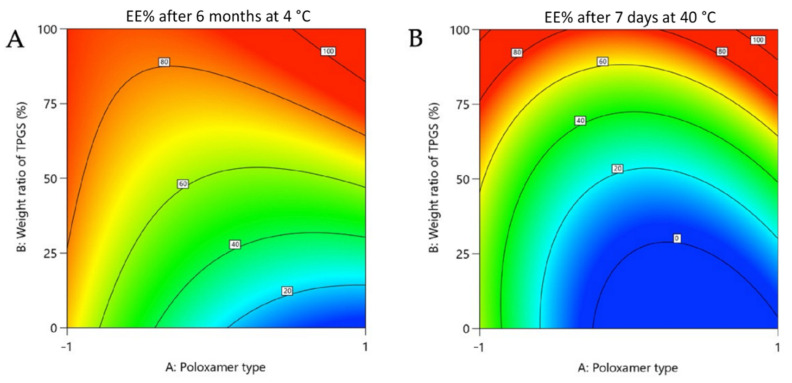
Contour plot showing relative effect of poloxamer type and weight ratio of TPGS on rapamycin EE% after 6 months at 4 °C (**A**) and 7 days at 40 °C (**B**).

**Figure 5 pharmaceutics-14-00569-f005:**
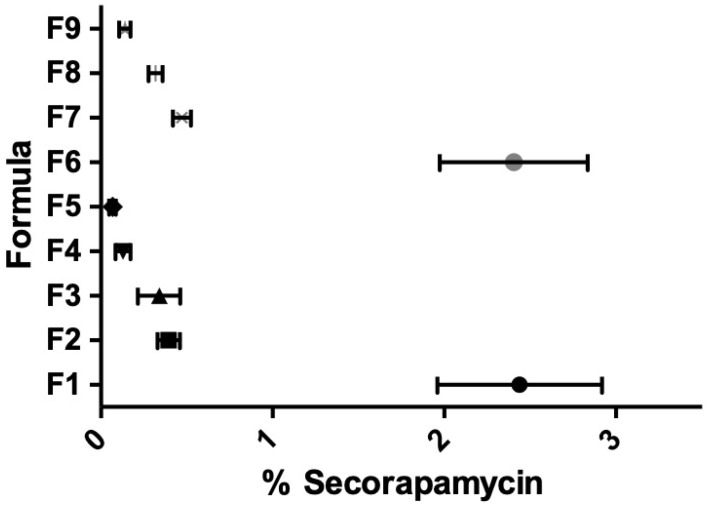
Percentage of secorapamycin, the main degradation product of rapamycin, according to the formulation studied as captioned and described in Table 1 (F1–F9).

**Figure 6 pharmaceutics-14-00569-f006:**
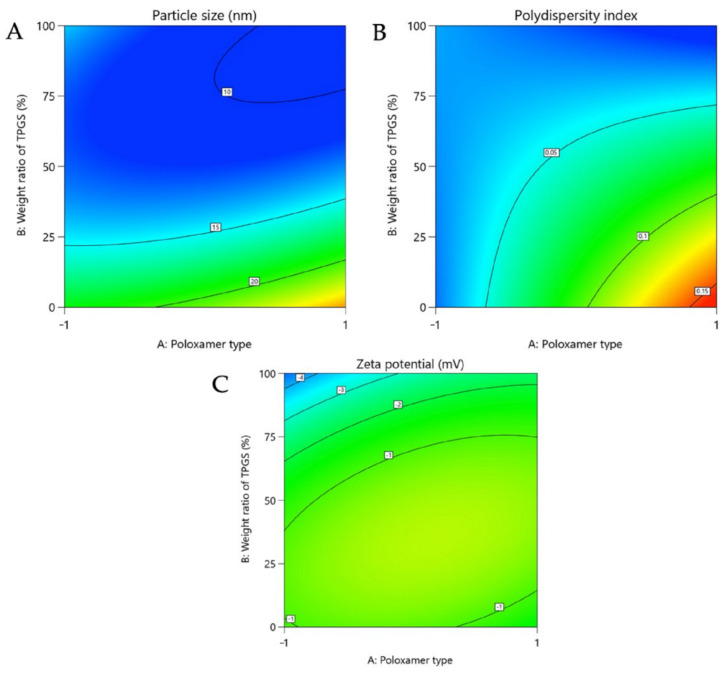
Contour plot showing relative effect of poloxamer type and weight ratio of TPGS on the size (**A**), the polydispersity index (**B**) and the zeta potential (**C**).

**Figure 7 pharmaceutics-14-00569-f007:**
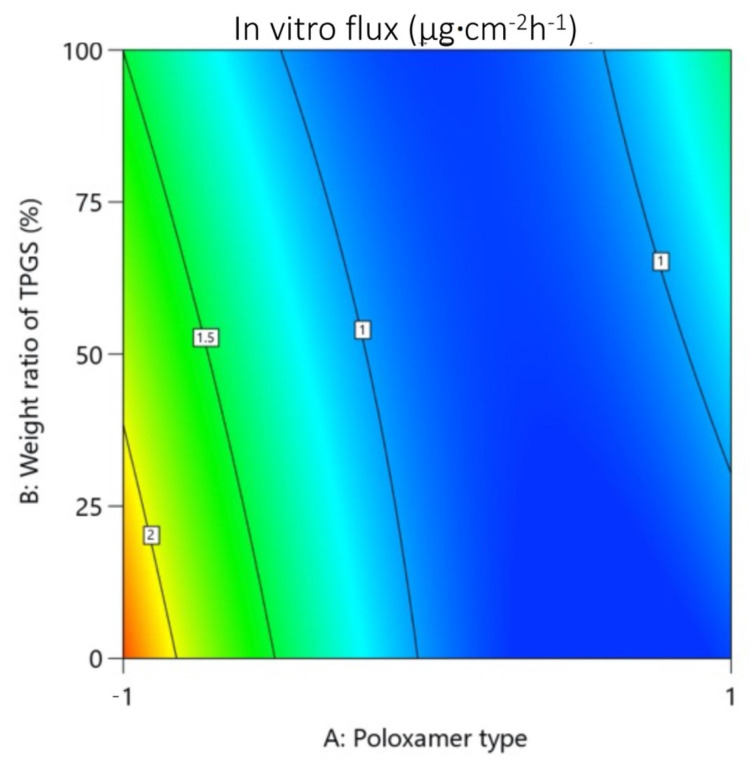
Contour plot showing relative effect of poloxamer type and weight ratio of TPGS on rapamycin in vitro flux.

**Figure 8 pharmaceutics-14-00569-f008:**
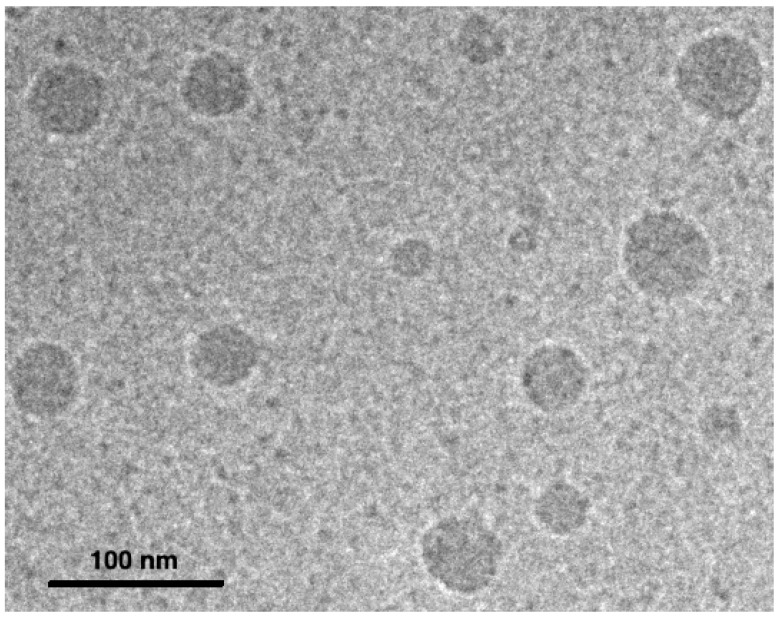
Cryo-TEM micrograph of micelles in the presence of rapamycin (formulation F3—2.5).

**Figure 9 pharmaceutics-14-00569-f009:**
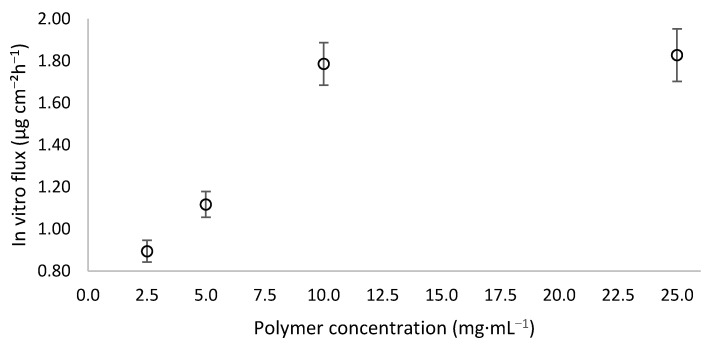
Impact of steady-state flux as a function of polymer concentration through Strat-M^®^ membrane. Average values were determined from 6 separate trials.

**Figure 10 pharmaceutics-14-00569-f010:**
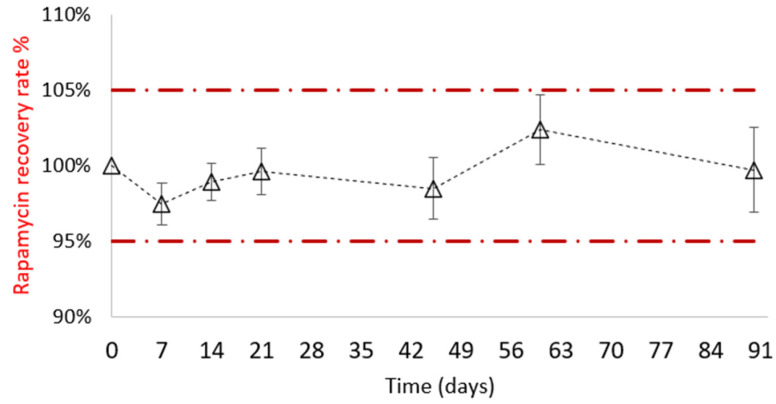
Rapamycin recovery rate % monitoring, calculated with respect to the initial time value (*n* = 3).

**Figure 11 pharmaceutics-14-00569-f011:**
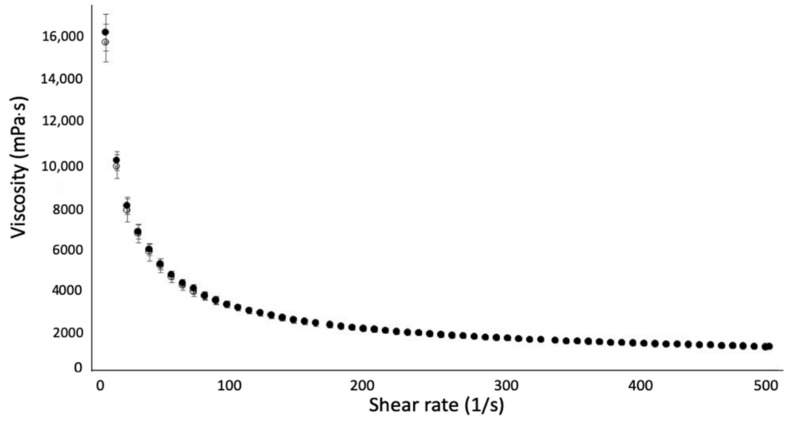
Rheological measurement over time of the rapamycin micellar hydrogel 0.1% (formulation F3) at day 0 (empty round) versus day 90 (full round).

**Figure 12 pharmaceutics-14-00569-f012:**
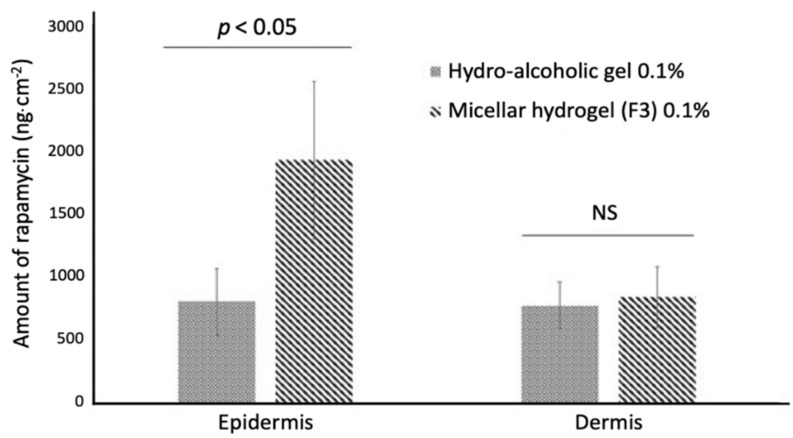
Cumulative amount of rapamycin per unit area (ng∙cm^−2^) over 24 h from micellar or hydroalcoholic gel through epidermis and dermis of human skin. Data are presented as the mean ± SD of 6 assays. NS: non-significant.

**Table 1 pharmaceutics-14-00569-t001:** Formulae, independent and dependent variables investigated in the full factorial design for rapamycin polymeric micelle preparation and the constraints applied for optimization.

Formulae (F_n_)	Factors (Independent Variables)	
	A: Poloxamer Type (−1: P123, 0: No Poloxamer, 1: F127)	B: Weight Ratio of TPGS to Total Copolymer (% *w*/*w*)
F_1_	−1	0
F_2_	−1	30
F_3_	−1	50
F_4_	−1	70
F_5_	0	100
F_6_	1	0
F_7_	1	30
F_8_	1	50
F_9_	1	70
**Responses (Y_n_, Dependent Variables)**		**Constraints**
Characterization	Y_1_: Entrapment efficiency (%)	Maximize
	Y_2_: Micellar size (nm)	Minimize
	Y_3_: Polydispersity index	Minimize
	Y_4_: Zeta potential (mV)	None
Rapamycin EE% (chemical stability)	Y_5_: 4 °C for 6 months	Maximize
	Y_6_: 40 °C for 7 days	Maximize
In vitro study	Y_7_: Percutaneous flux (µg∙cm^−^² h^−1^)	Maximize

**Table 2 pharmaceutics-14-00569-t002:** Studied parameter responses and modeling data based on a quadratic model.

Formulae	1: EE (%) ^a^	2: Size (nm) ^a^	3: PDI ^a^	4: ZP (mV) ^a^	5: Stability ^a^		6: In Vitro Flux (µg∙cm^−2^ h^−1^) ^a^
					EE (%) 4 °C— 6 Months	EE (%) 40 °C—7 Days	
F1	83.2 ± 2.1	17.7 ± 0.5	0.030 ± 0.013	−1.2 ± 0.9	78.7 ± 1.5	53.1 ± 0.9	2.07 ± 0.13
F2	89.3 ± 2.2	16.0 ± 0.4	0.021 ± 0.005	−1.4 ± 2.0	75.1 ± 2.4	53.5 ± 0.9	1.85 ± 0.04
F3	89.1 ± 2.7	12.6 ± 0.3	0.026 ± 0.013	−1.2 ± 0.8	81.0 ± 2.5	65.4 ± 1.6	1.83 ± 0.13
F4	87.6 ± 2.8	10.9 ± 0.2	0.034 ± 0.019	−2.2 ± 1.0	82.5 ± 2.5	73.1 ± 1.2	1.65 ± 0.29
F5	88.8 ± 2.3	10.7 ± 0.1	0.020 ± 0.011	−2.5 ± 1.5	84.3 ± 2.9	77.1 ± 2.7	0.85 ± 0.11
F6	82.1 ± 2.5	29.7 ± 2.6	0.158 ± 0.004	−1.5 ± 1.0	3.1 ± 1.6	0.5 ± 0.3	0.85 ± 0.04
F7	88.7 ± 2.4	14.9 ± 0.9	0.126 ± 0.015	−0.4 ± 0.5	28.0 ± 0.9	11.0 ± 0.3	0.96 ± 0.11
F8	88.6 ± 2.7	12.5 ± 0.6	0.083 ± 0.029	−0.8 ± 0.4	75.8 ± 2.4	50.2 ± 2.6	0.89 ± 0.07
F9	88.2 ± 2.4	11.7 ± 0.3	0.049 ± 0.009	−0.7 ± 0.6	78.3 ± 3.2	65.8 ± 2.3	0.86 ± 0.14
Fit Statistics							
Adjusted R²	0.1643	0.8974	0.9150	0.0731	0.9134	0.9567	0.9028
Predicted R²	0.0195	0.8660	0.9026	–0.3066	0.9000	0.9501	0.8685
Adequate precision	5.0306	20.2157	25.5917	0.0910	2.7519	2.1166	17.9242

^a^ All measurements are represented as mean ± standard deviation (*n* = 6). EE: entrapment efficiency, PDI: polydispersity index, ZP: zeta potential.

**Table 3 pharmaceutics-14-00569-t003:** Effect of polymer-to-drug ratio on micelle properties.

Formula	Polymers (TPGS + P123; TPGS/P123 50/50) Concentration (mg∙mL^−1^)	EE (%)	DL (%)	Size (nm)	PDI
F3–25	25	89.1 ± 2.7	3.4 ± 0.1	12.6 ± 0.3	0.026 ± 0.013
F3–10	10	89.1 ± 1.1	8.1 ± 0.1	13.9 ± 0.2	0.016 ± 0.090
F3–5	5	88.1 ± 1.9	14.7 ± 0.3	14.5 ± 0.2	0.015 ± 0.008
F3–2.5	2.5	88.4 ± 2.2	25.3 ± 0.6	84.7 ± 4.3	0.156 ± 0.010

## Data Availability

The data presented in this study are available in the article.

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
