# Peer review of "Mixed Polymeric Micelles for Rapamycin Skin Delivery"

_pharmaceutics, 2022, doi:10.3390/pharmaceutics14030569_

Round 1

Reviewer 1 Report

The manuscript by Le Guyader et al. deals with the topic of the development of the efficient delivery of rapamycin for the treatment of skin lesions in TSC patients. Authors proved that by utilizing specific micellar compositions, the stability of composition can be prolonged. I have some outstanding questions for the authors which should be addressed before acceptance in the Pharmaceutics journal.

  1. the abstract should not include the speculations about prolonged drug release and decrease of side effects when these parameters were not studied.
  2. Figure 5 is missing the legend about formulations which has been measured. Is hydro-alcoholic gel one of the compositions measured here? The stability (seco rapamycin formation) should be compared also for this composition if better stability for micelles is concluded.
  3. what is the safety profiles (e.g. cytotoxicity, skin sensitization, in vivo data) of used micellar compositions? Are concentrations of micelles in hydrogel relevant to both safety and efficient delivery of rapamycin? Please comment.
  4. what is the targeted tissue in FA? Usually, angiofibromas are formed from dermal tissue while the epidermis is uninvolved or atrophic. Figure 7 shows that the amount of rapamycin in the dermis is comparable between both tested compositions. The author speculates that the higher epidermal content can act as a depot for the dermis, but without this theory proven there might be no benefit in micellar composition treatment. Please comment on this aspect.
  5. Is there any information about the possibility of an intracellular (transcellular) route of rapamycin delivery when using micellar systems? Would this be beneficial for the efficiency of treatment? Or can rapamycin permeate easily through the membrane, so intracellular delivery is not necessary? Please comment

At the moment, I recommend the major revision of the manuscript. Authors should at least directly discuss in the manuscript the above-mentioned problems, in some cases, I would really recommend additional analyses to strengthen the conclusions in the manuscript.

Author Response

The manuscript by Le Guyader et al. deals with the topic of the development of the efficient delivery of rapamycin for the treatment of skin lesions in TSC patients. Authors proved that by utilizing specific micellar compositions, the stability of composition can be prolonged. I have some outstanding questions for the authors which should be addressed before acceptance in the Pharmaceutics journal.

  1. the abstract should not include the speculations about prolonged drug release and decrease of side effects when these parameters were not studied.

Our response:

These elements have been removed from the abstract and replaced by: “In addition, compared to hydroalcoholic gel formulations, the studied system allows for better biodistribution on human skin” (lines 31-32)

  1. Figure 5 is missing the legend about formulations which has been measured. Is hydro-alcoholic gel one of the compositions measured here? The stability (seco rapamycin formation) should be compared also for this composition if better stability for micelles is concluded.

Our response:

We measured secorapamycin in all studied formulations (F1-F9). F1-F9 are reported on the y-axis of Figure 5. The title of Figure 5 has been rewritten as follows: “Percentage of secorapamycin, the main degradation product of rapamycin, according to the formulation studied as captioned and depicted in Table 1 (F1-F9)“ (lines 346-347).

Regarding the second part of the commentary, we would like to stress that the hydroalcoholic gel was prepared extemporaneously to study only comparatively the permeation. That’s why stability studies were not performed on the hydroalcoholic gel. Nevertheless, we found that the higher the inclusion in the micelles (high EE%), the lower the content of degradation product (Figure 5), implying that the free part is more exposed to degradation and thus this would also be the case for rapamycin solubilized as is in the hydroalcoholic gel.

  1. what is the safety profiles (e.g. cytotoxicity, skin sensitization, in vivo data) of used micellar compositions? Are concentrations of micelles in hydrogel relevant to both safety and efficient delivery of rapamycin? Please comment.

Our response:

The studied micelles are composed of TPGS and/or poloxamers.

According to the handbook of excipients, poloxamers are regarded as nontoxic and nonirritant materials (handbook of excipients, Ed.6, p506). Poloxamers may therefore be used in ointments, suppository bases, and gels at concentrations ranging from 0,3 to 50 % w/w. In the present study, the gel with dispersed polymeric micelles contains no more than 12.5 mg of poloxamer per mL, i.e. no more than1.25 % w/w of poloxamer. TPGS as for it, is also present at the same level, which represent low levels compared to what has been commonly used in topical formulations [20-22].

Accordingly, we propose to add the following sentence: “both present in low concentrations of no more than 2.5 % w/w of polymers due to their very low CMC“ (lines 509-510).

  1. Eur J Pharm Sci. 2005;25(4–5):445–53;
  2. Pharm Res. 2006;23(2):243–55;
  3. Theranostics 2018;8(2):464-485

  1. what is the targeted tissue in FA? Usually, angiofibromas are formed from dermal tissue while the epidermis is uninvolved or atrophic. Figure 7 shows that the amount of rapamycin in the dermis is comparable between both tested compositions. The author speculates that the higher epidermal content can act as a depot for the dermis, but without this theory proven there might be no benefit in micellar composition treatment. Please comment on this aspect.

Our response:

Please see the response to comment 5 below

  1. Is there any information about the possibility of an intracellular (transcellular) route of rapamycin delivery when using micellar systems? Would this be beneficial for the efficiency of treatment? Or can rapamycin permeate easily through the membrane, so intracellular delivery is not necessary? Please comment

Response to Comments 4 and 5:

To effectively treat FA, rapamycin must be able to penetrate the stratum corneum to the dermis, the layer where angiofibromas are formed. The poor permeation of rapamycin (> 500 Da and 1<logP<3) has led to the development of many formulations to improve it and we have already comparatively evaluated their performance using both StratM® and human skin [Pharmaceutics 2020, 12, 1060].

As the hydroalcoholic gel formulation showed the best performance in terms of thermodynamic activity and permeation, it was used as a reference to evaluate that of our small-size polymeric micelles on human skin, and despite a much better bioaccumulation of rapamycin carried by the polymeric micelles, the concentration of the latter in the deeper layer was not different (Figure 7).

We propose to remove sentence “Probably, the excess part present in the epidermis can be therefore considered as a reservoir of active substance which will migrate into the deeper layer as the latter is exhausted of rapamycin“ and to add instead, the following new paragraph with new references to the manuscript:

“Based on these results and the fact that intercluster penetration (deposition between corneocytes and between corneocyte clusters) and/or within hair follicles have been suggested as the preferred *, but not the only [41-43], transport routes for polymeric micelles, we hypothesized that a concentration of micelles at the epidermal level, within which rapamycin is maintained stable chemically, may act as a depot for the dermis, which, upon repeated administration, could enhance the concentration of rapamycin in the deeper layer through continuous release of rapamycin resulting in the formation of a diffusion gradient. Nonetheless, further tests will have to be carried out to support these hypotheses“(lines 488-496 and lines 632-646).

[41-43]

Eur J Pharm Biopharm 2014;88:614-24;

J Control Release 2011;150:45-8;

Skin Pharmacol Physiol 2013;26:227-33

[44-46]

Materials 2020, 13, 366; doi:10.3390/ma13020366;

Chem. Soc. Rev. 2012, 41, 2718–2739;

Chem. Soc. Rev. 2017, 46, 4218–4244

At the moment, I recommend the major revision of the manuscript. Authors should at least directly discuss in the manuscript the above-mentioned problems, in some cases, I would really recommend additional analyses to strengthen the conclusions in the manuscript.

Reviewer 2 Report

  1. Figure 1, the author should also provide the IR spectrum of rapamycin only to compare the characteristic peaks, major peaks should be labeled and depict the relative structure.
  2. In 2.2, the author could also provide the elution time for rapamycin with the described HPLC method.
  3. In the introduction where the authors mentioned “ Due to their high solubility, high loading capacity and controllable release with 67 a long duration of action, biodegradable polymeric micelles have gained considerable 68 popularity [18]. Polymeric micelles are formed from copolymers, each consisting of hy- 69 drophilic and hydrophobic monomers, which self-assemble when the critical micelle con- 70 centration (CMC) is exceeded to form a specific core-corona structure.”, more related references talking about polymer-based micelle for drug encapsulation should be added, such as  1) Advanced Materials 17, no. 6 (2005): 657-669. (2) Angew. Chemie - Int. Ed. 2020, 59 (26), 10456–10460. (3)  Chem. Soc. Rev. 2015, 44(17), 6161–6186. 

Author Response

  1. Figure 1, the author should also provide the IR spectrum of rapamycin only to compare the characteristic peaks, major peaks should be labeled and depict the relative structure.

Our response:

Within the manuscript, we already depict the characteristic peaks of rapamycin (lines 244-260). Please see below what is highlighted in yellow:

“The ATR-FTIR spectrum of a non-loaded-rapamycin mixture prepared by extemporaneously dispersing/solubilizing rapamycin and TPGS/poloxamer P123 (1/1 w/w) in D2O (red line) shows a broad but weakly intense peak stretching from 3200 to 3550 cm-1, corresponding to OH vibrations with intermolecularly engaged H (Figure 1). In the case of rapamycin-loaded micelles, the same peak became much more intense, which would imply a strengthening of this phenomenon, probably due to stronger polar interactions between rapamycin and the hydrophile part of the polymers. Furthermore, the near disappearance of a cluster of peaks located between 2875 and 2932 cm-1 observed in the case of rapamycin-loaded micelles, most probably related to the C-CH3, O-CH3 and aliphatic C-H stretching vibrations and characteristic of certain functions of the active substance, seems to clearly indicate that the latter is deeply included in the inner part of the micelles, becoming non-visible to the FTIR detection. Likewise, the conjugated C=O and conjugated C=C stretching vibrations (1718 cm-1 and 1635 cm-1, respectively) are no longer present for the rapamycin-loaded micelles. In a nutshell, examination of the spectrum recorded for the rapamycin-loaded micelles gives a completely different configuration, showing the disappearance of the peaks pertaining to rapamycin, implying its deep inclusion within micelles“.

Nonetheless, we have added an FTIR spectrum of rapamycin as supplemental material at the end of the manuscript.

SI: FTIR spectrum of rapamycin

  1. In 2.2, the author could also provide the elution time for rapamycin with the described HPLC method.

Our response:

Retention times has been reported accordingly, i.e. about 20.0 min for rapamycin and 18.5 min for secorapamycin (lines 11-113).

  1. In the introduction where the authors mentioned “Due to their high solubility, high loading capacity and controllable release with a long duration of action, biodegradable polymeric micelles have gained considerable popularity [18]. Polymeric micelles are formed from copolymers, each consisting of hydrophilic and hydrophobic monomers, which self-assemble when the critical micelle concentration (CMC) is exceeded to form a specific core-corona structure.”, more related references talking about polymer-based micelle for drug encapsulation should be added, such as1) Advanced Materials 17, no. 6 (2005): 657-669. (2)  Chemie - Int. Ed. 202059 (26), 10456–10460. (3)  Chem. Soc. Rev. 201544(17), 6161–6186. 

Our response:

We have added these proposed references (lines 575-582).

Reviewer 3 Report

Mixed polymeric micelles for rapamycin skin delivery- Well executed research study and written manuscript. Few missing gaps.

  1. Authors missed writing materials in section 2.0
  2. If no constraints, Y4 zeta potential cannot be a Response to optimize.
  3. The authors should have done scanning electron microscopy to check the incorporation of micelles in the porous structure of hydrogels.
  4. Section 2.3.1- Line 127-129. Where is the data of ANOVA and model significance? The authors have directly shown Quadratic model data. Why only this model was used?
  5. How the formulation was optimized? Whether authors have used mathematical modeling? Just observing contour plots formulation cannot be optimized.
  6. Stability data should be compared with the initial data of that particular batch. How authors used stability data to decide the optimum composition?

Author Response

Mixed polymeric micelles for rapamycin skin delivery- Well executed research study and written manuscript. Few missing gaps.

  1. Authors missed writing materials in section 2.0

Our response:

All materials’ origins are reported throughout 2.1. Polymeric micelles and hydrogel preparation. Indeed, to avoid repetition in the manuscript and ease the reading of this part, we have chosen to mention the origin of each material parallel to the method used for polymeric micelles and hydrogel preparation.

  1. If no constraints, Y4 zeta potential cannot be a Response to optimize.

Our response:

We found via preliminary test data that the zeta potential is not influenced by the polymer composition in the studied experimental area, so we did not impose a constraint for this criterion.

  1. The authors should have done scanning electron microscopy to check the incorporation of micelles in the porous structure of hydrogels.

Our response:

We did not use SEM because controversial use of SEM has been pointed out in the context of swollen gels (Microscopic Structure of Swollen Hydrogels by Scanning Electron and Light Microscopies: Artifacts and Reality10.3390/polym12030578).

Section 2.3.1- Line 127-129. Where is the data of ANOVA and model significance? The authors have directly shown Quadratic model data. Why only this model was used?

Our response:

We have rephrased as follows: “The influence of parameters on the controlled criteria was modeled. Of the models tested, only quadratic least squares models led to a difference between adjusted and predicted R2 of no more than 0.2 [27], and this was observed for almost all responses (ANOVA, p < 0.05) studied apart from zeta potential (ANOVA, p = 0.3525)“ (lines 292-296).

  1. How was the formulation optimized? Whether authors have used mathematical modeling? Just observing contour plots formulation cannot be optimized.

Our response:

The modeling approach only allowed to visualize the interdependence of the parameters on formulation criteria. Many criteria have been studied and are not influenced in the same way by the studied parameters, therefore, it was eventually up to the experimenter to choose the optimal compositions, not mathematically, but according to the experimenter.

For instance (lines 433-436): “However, although the optimal polymer concentration is around 10 mg mL-1, we preferred to continue working at 25 mg mL-1 to give maximum stability to the micellar system to cope with a change of environment due to the formulation of a hydrogel containing 0.1 % of rapamycin.”

  1. Stability data should be compared with the initial data of that particular batch. How authors used stability data to decide the optimum composition?

Our response:

Has been added the following sentence to the manuscript: “The evolution of various parameters is compared over time, between formulations and with respect to time zero. In case these parameters are all equivalent at T0, relative differences between the formulations help identify the best polymer composition among those tested.” (lines 235-238).
